

# Non-bleached colonies of massive *Porites* may attract fishes for selective grazing during mass bleaching events

Eri Ikeuchi[1], Yoshikazu Ohno[2], Akira Iguchi[3] and Takashi Nakamura[4,5]

[1] Chemistry, Biology, and Marine Sciences Course, University of the Ryukyus, Nishihara, Okinawa, Japan
[2] Marine Genomics Unit, Okinawa Institute of Science and Technology Graduate University, Onna-son, Okinawa, Japan
[3] Department of Bioresources Engineering, National Institute of Technology, Okinawa College, Nago, Okinawa, Japan
[4] Department of Chemistry, Biology and Marine Science, Faculty of Science and Tropical Biosphere Research Center, University of the Ryukyus, Nishihara, Okinawa, Japan
[5] Japan Science and Technology Agency (JST)/Japan International Cooperation Agency (JICA) SATREPS, Tokyo, Japan

## ABSTRACT

In this study we investigated the variation in grazing scar densities between bleached and non-bleached colonies of massive *Porites* species in Sekisei Lagoon (Okinawa, southwestern Japan) during a mass bleaching event in 2016. The grazing scar densities and bleaching susceptibility varied among neighboring colonies of massive *Porites* spp. However, non-bleached colonies had significantly more surface scars than bleached colonies. One explanation for these variations is that corallivorous fishes may selectively graze on non-bleached, thermally tolerant colonies. This is the first report of a relationship between grazing scars and the bleaching status of massive *Porites* spp. colonies during a mass bleaching event.

Corresponding author
Eri Ikeuchi, e.ikeuchi73@gmail.com

## INTRODUCTION

Reef fishes belonging to the families Chaetodontidae, Labridae (*Bonaldo & Bellwood, 2011*), and Tetraodontidae (*Jayewardene et al., 2009*) make scars on the skeletons of corals when they graze on algae, epifauna and endofauna. It is suggested that grazing by parrotfishes in the Caribbean and on the Great Barrier Reef may have serious consequences for the dynamics of coral populations (*Bruckner & Bruckner, 1998*; *Mumby, 2009*; *Bonaldo & Bellwood, 2011*; *Bonaldo, Krajewski & Bellwood, 2011*; *Bonaldo, Welsh & Bellwood, 2012*; *Bonaldo, Hoey & Bellwood, 2014*; *Cole et al., 2011*). As the main target coral species for fish grazing, massive *Porites* species are the representative corals.

Six massive *Porites* species are known from marine waters of southwestern Japan (*Nishihira & Veron, 1995*). Many poritid species that have a massive colony morphology are known for their stress tolerance (*Loya et al., 2001*) and longevity, which may enable them to form relatively large colonies (several meters in diameter) compared with other coral species (*Veron, 2000*). Therefore, massive *Porites* species are thought to be ecologically important

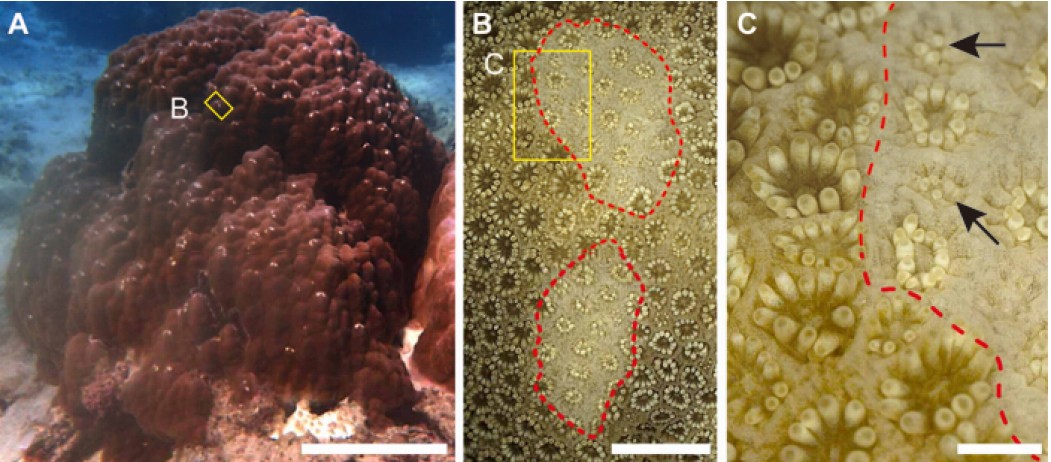

**Figure 1  Grazing scars on the surface of massive *Porites*.** (A) The yellow box shows the part of the surface enlarged as the image in (B). Scale bar: 20 cm. (B) Microscopic image of the part of the *Porites* corresponding to the yellow box in (A). Red dotted lines show the area of the grazing scar. The yellow box area indicates the area enlarged as the image in (C). Scale bar: 5 mm. (C) Marginal area of the edge of the scar in (B). Black arrows indicate light-colored polyps under regeneration. Scale bar: 1 mm.

reef builders (*Iguchi et al., 2014*), and are often used for estimating past environmental conditions, including temperature (e.g., *Gagan et al., 2000*).

In shallow coral reef habitats around Okinawa Island (southwestern Japan), feeding scars are commonly observed on the surface of massive *Porites* colonies (Fig. 1). The scar densities vary among neighboring colonies, but the reasons for this variability have not been explored. Direct damage to live tissues caused by grazing can lead to serious problems for corals, as heavily damaged colonies die (e.g., *Treeck & Schuhmacher, 1997*), and this process contributes to the dynamics of coral populations. Furthermore, scars on massive *Porites* colonies may provide suitable settlement sites for macro-borers including vermetid gastropods (*Dendropoma maximum*) and Christmas tree worms (e.g., *Spirobranchus giganteus*) (*Nishihira, 1996*). Therefore, clarifying why grazing scar density variations occur on the surface of *Porites* spp. is important in determining the ecological processes affecting populations of these corals, and for understanding the establishment and maintenance of microscale biodiversity around *Porites* colonies.

Coral bleaching is one of the threats for degradation of coral reef ecosystems and caused mainly by thermal stress, followed by the breakdown of the symbiotic relationship between a coral and its symbiotic zooxanthellae (*Hoegh-Guldberg, 1999*). However, sympatric colonies often show variability in bleaching susceptibility (e.g., *Jones et al., 2008*), which is partially explained by differences in the types of zooxanthellae in the host tissues, particularly differences in stress tolerance between zooxanthellae of clades C and D (e.g., *Baker, 2003*). During summer in 2016, mass coral bleaching occurred in Sekisei Lagoon (southwestern Japan); it involved >95% of colonies, which bleached as a consequence of prolonged high seawater temperatures. It is reported that seawater temperatures in Sekisei Lagoon between June and September 2016 were over 30 °C (*Ministry of the Environment, 2017*). Among many bleached corals at 11 surveyed sites within an area of <100 m radius

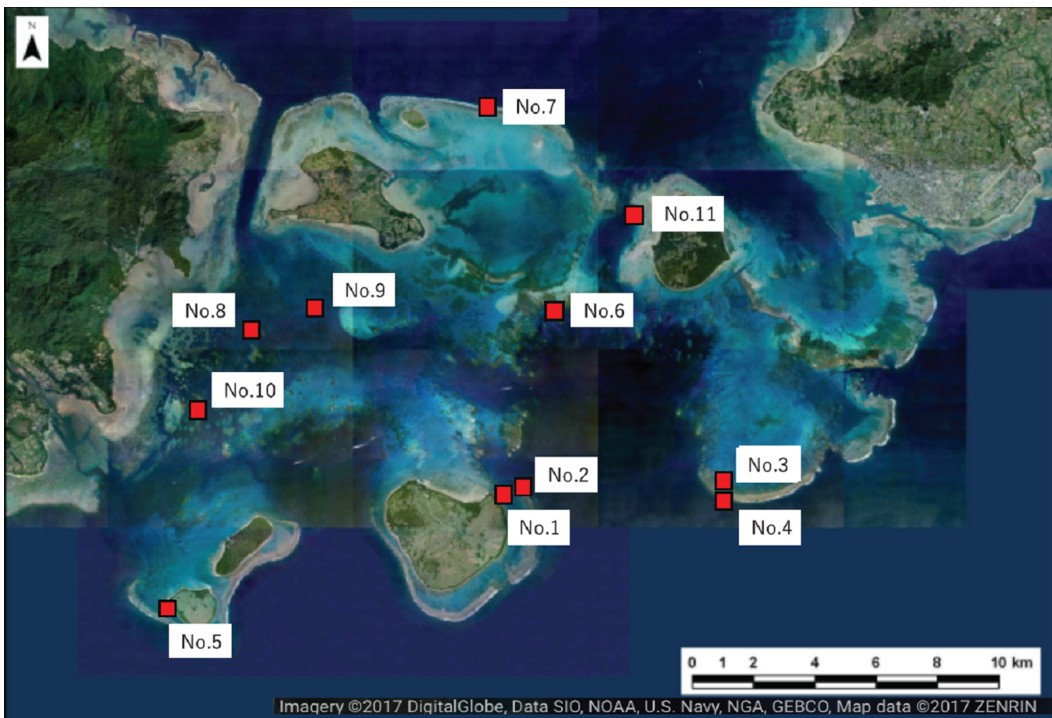

**Figure 2** **Location of the study sites within Sekisei Lagoon, Okinawa, Japan.** Map of Sekisei Lagoon and location of 11 survey sites.

in the lagoon, we observed numerous bleached and non-bleached massive *Porites* colonies. At these sites we conducted scar density surveys of massive *Porites* species to investigate the influence of bleaching on fish grazing on these corals.

## MATERIALS AND METHODS

We selected 11 sites for the survey of massive *Porites* species (mainly *Porites australiensis,* *P. lobata,* and *P. lutea*) in Sekisei Lagoon (Fig. 2 and Table 1). The survey was carried out during 3–12 September 2016. Prior to taking place, the survey was approved by the Ministry of the Environment.

Underwater observations were conducted at each site using SCUBA. We recorded the following parameters for all massive *Porites* colonies found during a 30 min diving survey at each site: (i) the degree of scarring (5 stages: 0–4; see Fig. 3 for details); (ii) the maximum colony diameter as a measure of colony size, recorded in one of 5 categories (1: <30 cm; 2: 31–50 cm; 3: 51–80 cm; 4: 81–110 cm; 5: >111 cm); (iii) depth (m); and (iv) the occurrence of bleaching (bleached vs. non-bleached). In our preliminary survey, we calculated the degree of scarring as the area of grazing scars per that of the colony surface (Table S1). Based on the criteria, we judged the degree of scarring by visual inspection. We defined completely bleached colony not covered by algae as bleached one in this study. Thus, the colonies which we observed seemed to be alive because the dead colonies are soon covered by algae. We removed partially bleached colonies from our analysis. The grazing scars were clearly visible in the field, even on bleached colonies (Figs. 3E and 3F). For several grazed

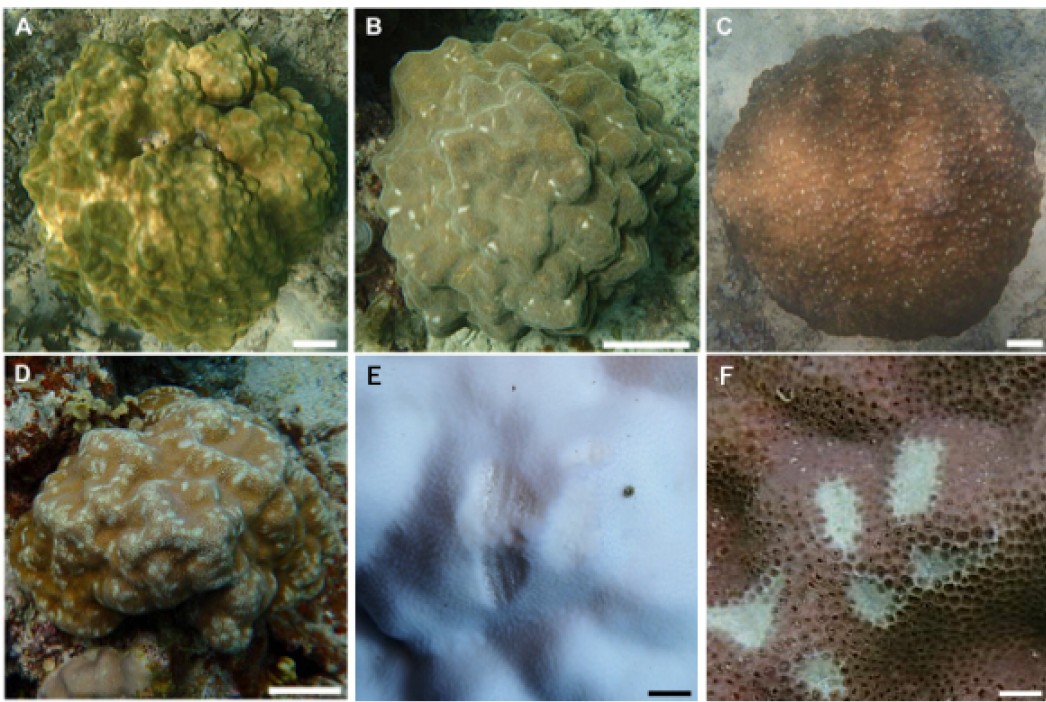

**Figure 3 Images showing the degree of scarring on massive *Porites* in the field.** (A) <5%; (B) 6–15%; (C) 16–25%; (D) >26%; scale bars A–D: 10 cm, E and F: 1 cm. Magnified images of a grazing scar on bleached (E) and non-bleached (F) colonies in the field.

**Table 1 Details of study sites.**

| No. | Latitude | Longitude | Depth (m) | No. of bleached colonies | No. of non-bleached colonies | Total no. of colonies |
|---|---|---|---|---|---|---|
| 1 | E24°14′54.5″ | N124°01′50.0″ | 7.6 ± 1.5 | 6 | 9 | 15 |
| 2 | E24°15′01.1″ | N124°02′05.4″ | 6.4 ± 0.6 | 11 | 19 | 30 |
| 3 | E24°15′04.1″ | N124°06′03.0″ | 4.3 ± 1.3 | 10 | 20 | 30 |
| 4 | E24°14′50.8″ | N124°06′04.4″ | 13.7 ± 2.4 | 14 | 6 | 20 |
| 5 | E24°12′45.5″ | N123°55′14.7″ | 5.4 ± 1.5 | 12 | 26 | 38 |
| 6 | E24°18′17.6″ | N124°02′46.9″ | 8.3 ± 1.8 | 9 | 21 | 30 |
| 7 | E24°22′02.5″ | N124°01′27.0″ | 13.4 ± 1.4 | 12 | 18 | 30 |
| 8 | E24°17′55.8″ | N123°56′51.0″ | 4.5 ± 2.4 | 2 | 8 | 10 |
| 9 | E24°18′18.4″ | N123°58′04.3″ | 7.2 ± 2.6 | 8 | 9 | 17 |
| 10 | E24°16′25.5″ | N123°55′51.2″ | 10.5 ± 0.5 | 7 | 23 | 30 |
| 11 | E24°20′00.0″ | N124°04′18.3″ | 4.3 ± 0.9 | 7 | 9 | 16 |

colonies we took high-magnification images of the grazing scars using a Keyence VHX-1000 digital microscope (Osaka, Japan). We also took digital images of all *Porites* colonies using a digital camera (TG-3; OLYMPUS, Tokyo, Japan) in an underwater housing (PT-056; OLYMPUS).
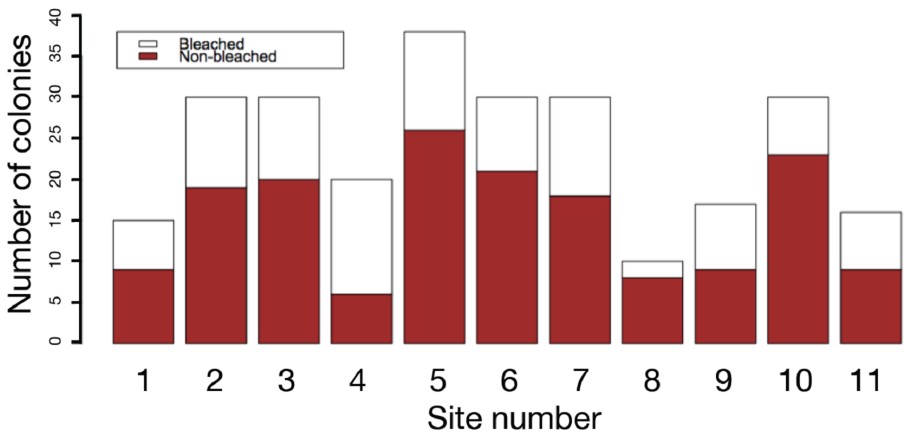

**Figure 4** **Frequency of bleached and non-bleached colonies of massive *Porites* recorded at each site.** *X* axis: site number; *Y* axis: number of colonies.

To investigate the occurrence of significant relationships among colony size, degree of scarring, and the depth of bleached and non-bleached colonies, we applied the Mann–Whitney $U$-test using software R ver. 3.2.4 (*R Core Team, 2016*). We used an ordered logistic regression model (a cumulative link mixed model) in which the response variable was the degree of scarring, and the explanatory variables were colony size, depth, bleaching occurrence, and site. This analysis was also performed using software R ver. 3.2.4 (*R Core Team, 2016*) and ordinal package (*Christensen, 2015*).

## RESULTS AND DISCUSSION

The results of field surveys of massive *Porites* species at the 11 sites within Sekisei Lagoon showed that 37% of the surveyed colonies were bleached (total number of observed colonies: 266; Table 1). The percentages of each degree of colony size are as follows; 1: 15.1%; 2: 54.1%; 3: 20.8%; 4: 7.5%; 5: 2.5%. Both bleached and non-bleached colonies were recorded at all sites (Fig. 4), but the number of grazing scars on non-bleached colonies was 2.88 times greater than that on bleached colonies (Fig. 5A; Mann–Whitney test, $p < 0.01$). We found no significant effect of colony size or habitat depth on the occurrence of bleaching (Figs. 5B and Fig. 5C; Mann–Whitney test, $p > 0.1$). The ordered logistic regression model analysis showed that the occurrence of bleaching was the explanatory variable that was significantly correlated with the degree of scarring ($Z$-score $= -7.406$; $p < 0.01$). In the case of site, no. 7 was also significant ($Z$-score $= 2.177$, $p = 0.03$) but other sites not. Colony size and depth were not significant explanatory variables ($Z$-scores $= -0.268$ and $-0.864$, respectively; both $p$ values $> 0.1$).

Although we did not identify the cause of these variations, this is the first report suggesting a relationship between bleaching occurrence and grazing scar densities on massive colonies of *Porites* spp. during the period of a mass bleaching event. There are several possible explanations for the large variations in grazing scar densities on massive *Porites* colonies in the field. Firstly, corallivorous fishes may selectively choose colonies based on nutrition. For example, it is thought that the grazing scars caused by parrotfishes occur as they feed

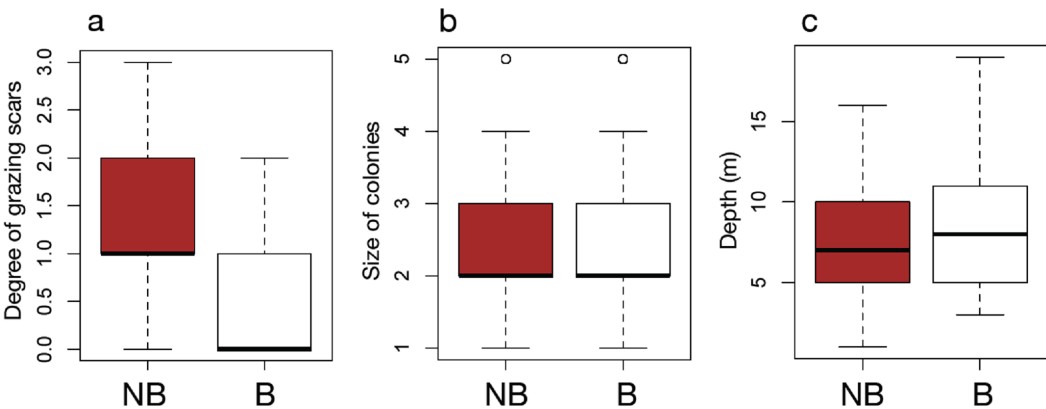

**Figure 5** **Box plots of each factor between non-bleached and bleached colonies of massive *Porites*.** Box plots of the degree of scarring (A), size of colonies (B), and depth (C) between non-bleached (NB) and bleached (B) colonies of massive *Porites*.

on coral tissues, but also on macro-borers (*Rotjan & Lewis, 2005*), from which they obtain additional nutrients not readily provided by herbivory. Thus, corallivorous fishes may select non-bleached colonies to obtain certain nutrients in addition to the energy derived from coral tissues (*Rotjan & Lewis, 2009*).

Furthermore, corallivorous fishes may selectively prey on coral colonies having thicker tissues, that may be originally tolerant to thermal stress (*Loya et al., 2001*), as potentially nutritious food sources. *Lough & Barnes (2000)* reported that the tissue thickness of massive *Porites* colonies varied among colonies and sites on the Great Barrier Reef. Reproductive stage of the coral may also attract grazer fishes. *Rotjan & Lewis (2009)* suggested that parrotfishes may selectively prey on the tissue of colonies of *Montastrea annuralis* containing mature eggs, because of its higher nutritional value. As massive *Porites* species are dioecious (*Harriott, 1983*), the scar densities may be variable among male and female colonies. Further studies are required to assess whether differences in coral tissue thickness can be detected by coral predators.

The various scar densities among colonies could be produced by random grazing of fishes. As the growth rates of massive *Porites* vary among colonies (*Iguchi et al., 2012*; *Hayashi et al., 2013*), the speed of healing of grazing scars on the colony surface is also likely to be highly variable among colonies. Massive *Porites* species are reported to suspend growth during bleaching events (*Suzuki et al., 2003*). In fast-growing colonies, the scars may rapidly be covered by newly formed live tissue, and as a result the colonies could appear to have fewer scars. On the other hand, in colonies having slow growth rates the scars may not heal rapidly, and their number on the surface could appear to be greater. In this context, the bleached colonies in Sekisei Lagoon should have had a much greater scar density than non-bleached colonies because of reduced growth (healing) during the bleaching period.

The imbalance between grazing frequency and/or intensity as a function of healing speed (colony growth) should determine the appearance of colonies, but the low scar densities observed for bleached colonies in Sekisei Lagoon cannot be explained by variations in

the speed of scar healing. Therefore, we inferred that fishes selectively graze on non-bleached massive *Porites* colonies (or those less likely to be bleached). Future studies should investigate tissue thickness among grazed and non-grazed colonies, and reciprocal transplantation experiments using fragments of highly-grazed and non-grazed colonies should be performed to study the mechanisms underlying the temporal and spatial variations in scar densities on the surface of massive *Porites* colonies.

## ACKNOWLEDGEMENTS

We thank Mitsuhiro Ueno and Masahiko Sunagawa for their support during the field survey. We also thank Kana Kojima and Mariyam Shidha Afzal for their assistance in the field and their insightful comments and suggestions.

### Funding

This work was supported by the Japan Science and Technology Agency (JST)/Japan International Cooperation Agency (JICA) SATREPS. The funders had no role in study design, data collection and analysis, decision to publish, or preparation of the manuscript.

### Grant Disclosures

The following grant information was disclosed by the authors:
Japan Science and Technology Agency (JST)/Japan International Cooperation Agency (JICA) SATREPS.

### Competing Interests

The authors declare there are no competing interests. Takashi Nakamura is a chief advisor for the project under the umbrella of Japan Science and Technology Agency (JST)/ Japan International Cooperation Agency (JICA) SATREPS, but not an employee.

### Author Contributions

- Eri Ikeuchi conceived and designed the experiments, performed the experiments, wrote the paper, prepared figures and/or tables, reviewed drafts of the paper.
- Yoshikazu Ohno analyzed the data, prepared figures and/or tables, reviewed drafts of the paper.
- Akira Iguchi analyzed the data, wrote the paper, prepared figures and/or tables, reviewed drafts of the paper.
- Takashi Nakamura contributed reagents/materials/analysis tools, wrote the paper, reviewed drafts of the paper.

### Data Availability

The raw data has been supplied as a Supplemental File.

## Supplemental Information

Supplemental information for this article can be found online at http://dx.doi.org/10.7717/peerj.3470#supplemental-information.

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
