# Peer review of "Non-bleached colonies of massive Porites may attract fishes for selective grazing during mass bleaching events"

_PeerJ, doi:10.7717/peerj.3470_

## Round 0.1 · original submission · Minor Revisions

This is an interesting, well written and worthwhile paper, however both I and the reviewers would like to see more comprehensive analyses of your results (see reviews and annotated pdf). In particular I think you need to account for possible interaction of species, site etc using some sort of mixed model. I have found the book: Data Analysis with R Statistical Software by Rob Thomas et al to be a very helpful, no nonsense guide to analysing data such as you have.

Reviewer 1 ·

Basic reporting

On line 146, independent paragraph is not needed because authors discuss the same topic (relation between grazing scar densities and coral healing speed) as do in the above paragraph.

Experimental design

I am curious about the definition of bleached coral in this manuscript. Does bleached coral include both alive and dead colonies? both completely bleached colonies and partially bleached colonies? This definition affects discussion in which the thickness of tissues is considered as an important factor. Clear definition is needed in Materials and Methods.

Statistical analyses are not proper. I am not statistician so just note deficiencies.
As figure 4 shows, authors surveyed 11 sites, but compared number of grazing scars (additionally colony size and habitat depth) only between bleached and non-bleached by Mann-Whitney U-test (see lines 99, 108-112). Site and interaction between site*bleached/non-bleached are better to be included as explanatory variables.

Authors used an ordered logistic regression model with the degree of scarring as the response variable, and colony size, depth, and bleaching occurrence as explanatory variables. In this analysis, site is ignored. Samples (coral colonies) in a site are not independent as corals in different sites, and therefore site is better to be included as an explanatory variable. Additionally, descriptions about statistical values and R package (and its version) are necessary.

Validity of the findings

no comment

Additional comments

Authors surveyed the grazing scars on bleached and non-bleached Porites in 11 sites in Sekisei lagoon during the mass-bleaching event and found that grazing scars were more abundant on non-bleached corals than on bleached corals. This finding is valuable and worth being published in PeerJ. I just commented on statistical analyses to strengthen their achievement.

·

Basic reporting

no comment

Experimental design

no comment

Validity of the findings

no comment

Additional comments

This paper clearly shows that non-bleached colonies of Porites spp. were grazed more frequently than bleached ones for the first time.

I have only three minor comments.

1. For the degree of scaring (5 stages: Line 89), % is shown in the caption of Figure 3 (line 173–174). Please explain how to calculate the percentage. Also describe how to relate to the degree as is shown for the 5 categories of colony size (Line 90–91). For example, degree 0: 0%; 1: < 5%, etc.

2. The grazing scars are not so clearly visible on the bleached colony shown in Fig. 3e. The picture should be exchanged.

3. For the reason why non-bleached colonies were selectively grazed (Line 118–), I have one question. Is it possible that non-bleached colonies are preferred because they contain many zooxanthellae that will be digested by corallivorous fishes?

---

## Round 0.2 · accepted · Accept

You have addressed the comments from reviewers to my satisfaction. I take a small issue with the term "marginally significant" though (line 128). If you are working to the p < 0.05 paradigm, things are either significant or they are not!